# The Chymase Mouse Mast Cell Protease-4 Regulates Intestinal Cytokine Expression in Mature Adult Mice Infected with *Giardia intestinalis*

**DOI:** 10.3390/cells9040925

**Published:** 2020-04-09

**Authors:** Zhiqiang Li, Dimitra Peirasmaki, Staffan Svärd, Magnus Åbrink

**Affiliations:** 1Department of Biomedical Sciences and Veterinary Public Health, Swedish University of Agricultural Sciences, SE-75007 Uppsala, Sweden; zhiqiang.li@slu.se; 2Department of Cell and Molecular Biology, Uppsala University, SE-75124 Uppsala, Sweden; dimitra.peirasmaki@icm.uu.se

**Keywords:** mast cell, chymase, mMCP-4, infection, *Giardia intestinalis*

## Abstract

Mast cells have been shown to affect the control of infections with the protozoan parasite *Giardia intestinalis*. Recently, we demonstrated that *Giardia* excretory-secretory proteins inhibited the activity of the connective tissue mast cell-specific protease chymase. To study the potential role of the chymase mouse mast cell protease (mMCP)-4 during infections with *Giardia,* mMCP-4^+/+^ and mMCP-4^−/−^ littermate mice were gavage-infected with *G. intestinalis* trophozoites of the human assemblage B isolate GS. No significant changes in weight gain was observed in infected young (≈10 weeks old) mMCP-4^−/−^ and mMCP-4^+/+^ littermate mice. In contrast, infections of mature adult mice (>18 weeks old) caused significant weight loss as compared to uninfected control mice. We detected a more rapid weight loss in mMCP-4^−/−^ mice as compared to littermate mMCP-4^+/+^ mice. Submucosal mast cell and granulocyte counts in jejunum increased in the infected adult mMCP-4^−/−^ and mMCP-4^+/+^ mice. This increase was correlated with an augmented intestinal trypsin-like and chymotrypsin-like activity, but the myeloperoxidase activity was constant. Infected mice showed a significantly lower intestinal neutrophil elastase (NE) activity, and in vitro, soluble *Giardia* proteins inhibited human recombinant NE. Serum levels of IL-6 were significantly increased eight and 13 days post infection (dpi), while intestinal IL-6 levels showed a trend to significant increase 8 dpi. Strikingly, the lack of mMCP-4 resulted in significantly less intestinal transcriptional upregulation of IL-6, TNF-α, IL-25, CXCL2, IL-2, IL-4, IL-5, and IL-10 in the *Giardia*-infected mature adult mice, suggesting that chymase may play a regulatory role in intestinal cytokine responses.

## 1. Introduction

*Giardia intestinalis* (also named *G. lamblia* or *G. duodenalis*) is a non-invasive protozoan intestinal parasite found worldwide that mainly causes a self-limiting diarrheal-disease, i.e., giardiasis in humans and other mammals [1]. *Giardia*-infections are often asymptomatic but can also result in acute or chronic diarrhea, malabsorption and weight loss [2]. The parasite has been estimated to contribute to 200 million symptomatic infections per year [3] and since 1954, at least 132 water-borne outbreaks of giardiasis have been reported worldwide [4]. The *G. intestinalis* group is genetically diverse with eight described genotypes or assemblages, but only parasites from assemblage A and B infect humans [1]. Recent data show that *Giardia* is a significant factor in the induction of reduced weight gain and stunting of young children in low-resource settings [5,6]. Malnutrition due to *Giardia*-infections has also been replicated in mouse models [7,8]. The elderly is also a group that is more affected by intestinal infections, including infections with *G. intestinalis* [9,10,11]. However, there is little insight into how *Giardia spp*. cause disease; they are not invasive and secrete no known toxins [12]. Recent research suggests that *G. intestinalis* can secrete a large number of immunomodulatory proteins, possibly regulating host immune responses [13,14,15,16]. However, the mechanisms on how interactions between the host and *Giardia* either lead to parasite clearance or to disease remain to be understood.

Recent studies have shown the importance of different immune cells in giardiasis, where both innate and adaptive immunity seem to play significant roles [17,18,19]. Accumulated data suggest that there is a mixed Th1/Th2/Th17 response during giardiasis [19,20]. When *G. intestinalis* attach to the microvillus brush border of intestinal epithelial cells (IECs) there is a production of chemokines and cytokines that will attract immune cells to the intestinal submucosa [20,21,22]. However, the effects differ depending on model systems used. In cultured human IECs challenged by *G. intestinalis* trophozoites (assemblage B, isolate GS), several chemokines were highly up-regulated early—at 1.5 h after challenge [21]. In experimental infections of gerbils with the WB isolate (ATCC 50803) several chemokines and cytokines was up-regulated [20], whereas no major up-regulation of chemokine or cytokine genes were seen in 5–6-week-old female mice infected with trophozoites of the GS isolate [22]. Instead, the infection caused significant up-regulation of mast cell-specific proteases [22]. Significant numbers of mast cells are recruited to the small intestine during infection with *Giardia*, both to the mucosa and to the underlying submucosal connective tissue [23,24,25]. Mast cell deficient c-kit^w/wv^ (c-kit dependent deficiency in the mast cell compartment) mice and c-kit depleted mice both failed to control a *Giardia* infection [26], suggesting that mast cells and c-kit dependent mechanisms are necessary for elimination of a *G. intestinalis* infection. In addition, the complement factor 3a receptor was found to be important for recruitment of mast cells to the mucosa during *Giardia*-infections in mice [27]. There was also an effect on the adaptive immune system since mast cell-deficient mice failed to produce parasite-specific IgA [26]. Soluble *Giardia* trophozoite proteins can activate mast cells, and the secreted protein arginine deaminase (ADI) induces release of IL-6 and TNF-α [28], two cytokines that are important for clearance of *G. intestinalis* in mice [29,30,31].

The mouse mast cell-specific chymase, mouse mast cell protease (mMCP)-4, which is released by activated connective tissue mast cells, may degrade IL-6 and TNF-α to inhibit excessive inflammation [32,33]. mMCP-4 can regulate the intestinal barrier function by affecting tight junctions and smooth muscle cells lining the intestine [34]. Mast cell degranulation during *Giardia*-infections has been linked to intestinal hypermotility, one common symptom of giardiasis [35]. During other parasitic infections, mast cells degranulate and release a number of specific proteases, e.g., in rats infected with *Schistosoma mansoni*, the chymase rat mast cell protease-2 is systemically released into the blood by the majority of the hepatic mast cells during the period of parasite elimination [36]. In BALB/c mice, the mucosal mast cell chymases mMCP-1 and mMCP-2 are substantially increased upon intestinal nematode infection [37,38]. In addition, increased numbers of chymase-positive mast cells were found in the esophagus of patients with *Trypanosoma cruzi* infection [39]. However, these studies suggest that the mast cell-specific proteases may play important roles during parasitic infections, but most of these studies have used young (<10 weeks old) mice, i.e., mice that are still growing and gaining weight, while mature adult (>18 weeks old) mice are rarely used. It has also been shown that ageing is associated with structural and functional defects in the gut, including thickness of the mucus layer, diversity of the microbiota and immune mechanisms [11,40]. Thus, to investigate the potential role of the chymase mMCP-4 during experimental infections with *Giardia intestinalis* in mature adult mice, we here examined the intestinal immune responses in mature adult mMCP-4^+/+^ and mMCP-4^−/−^ littermate mice. Weight changes were recorded for eight or 13 days, and intestinal morphology with mast cell and granulocyte counts, trypsin-like, chymotrypsin-like, myeloperoxidase and neutrophil elastase activities, as well as intestinal cytokine and chemokine levels were evaluated in the mMCP-4^−/−^ and the mMCP-4^+/+^ mice. Our data suggests that the chymase mMCP-4 plays a regulatory role in the intestinal inflammatory responses in mature adult mice during infection with *G. intestinalis*.

## 2. Materials and Methods

### 2.1. Preparation of Giardia Trophozoites

*Giardia intestinalis, GS (clone H7),* belonging to assemblage B [1], was used for the experimental infection. The GS isolate (ATCC 50581) is a human isolate from Alaska, USA that has been used in experimental human infections [41]. The *G. intestinalis* trophozoites were cultured at 37 °C in polystyrene screw cap tubes in 10 mL of TYDK media supplemented with 10% heat inactivated bovine serum (Gibco, Thermo Fisher Scientific, Waltham, MA, USA), 10% sterile bile (12.5 mg/mL) and 1% Ferric ammonium citrate solution (2.2 mg/mL) with the final pH adjusted to 6.8 as described [42]. All TYDK medium components were purchased from Sigma-Aldrich (St. Louis, MO, USA), unless stated otherwise. *Giardia* GS trophozoites were pelleted at 931× *g* for 10 min after being kept on ice for 15 min, and then re-suspended in aliquots of cold phosphate buffered saline (PBS) at 10^6^ parasites per 100 μL, i.e., the infection dose for each mouse.

To get soluble *Giardia* proteins (sGPs), *G. intestinalis* GS trophozoites were chilled on ice for 15 min, washed three times with cold PBS and finally re-suspended in PBS. The trophozoites were sonicated three times for 30 s at 50 Watts and centrifuged at 931× *g* at 4 °C for 15 min to remove cell debris. The supernatants containing sGPs were kept at −20 °C until used. *Giardia* excretory-secretory proteins (ESPs) were obtained as previously described [15]. 

### 2.2. Mouse Breeding and Ethical Statement

The mouse mast cell protease-4 (mMCP-4) knockout mouse strain, congenic on the C57BL/6J Taconic genetic background (generation *n* ≥ 20), was kept under specific pathogen-free conditions at the Faculty of Veterinary Medicine and Animal Science, SLU, Uppsala, Sweden. Heterozygote littermate female and male mice were mated to produce the experimental infection groups. All experimental infections were conducted in agreement with the Swedish Animal Welfare Act and granted permission (#C140/15) from the Ethical Committee for Animal Experiments, Uppsala District Court. A maximum of five mice per cage were housed in individually ventilated cages (IVCs) (ca 501 cm^2^ Macrolon IIL cages, Techniplast Green Line GM500, Buguggiate, Italy) with aspen shavings for bedding, paper for nesting and a small house of paper. A 12 h light cycle was used, and water and rodent chow were provided ad libitum. No antibiotics were included in the diet, and the mice were never treated with any drugs.

### 2.3. Infections and Scoring of Mice

We infected mature adult (in the age range of 18 to 36 weeks, >18 weeks old) and young (in the age range of seven to 13 weeks, ≈10 weeks old) congenic mMCP-4^+/+^, mMCP-4^+/−^ and mMCP-4^−/−^ littermate mice by oral gavage with 10^6^
*Giardia* trophozoites in 100 µL PBS, in seven independent experimental infections (see Table 1). As a control of the growth performance in infected mice blinded congenic groups of mature adults and young noninfected littermate mMCP-4^+/+^, mMCP-4^+/−^ and mMCP-4^−/−^ mice were followed to day 13. Clinical scoring and weighing were performed in a blinded fashion, i.e., genotypes of littermate mice were not known to the assessor until determined at the experimental endpoint. The weights of the mice were recorded before infection (day 0) and then every second or third day post the *Giardia* infection until infected mice were euthanized at the experimental endpoint. Feces were collected every second or third day for *Giardia* detection. Blood was sampled at the experimental endpoint and allowed to clot to collect the serum. Tail tissue was collected for mMCP-4 genotyping. Small intestine (jejunum and duodenum) were collected and divided into two to three centimeter pieces and stored at −80 °C until used.

### 2.4. Genotyping of the mMCP-4 Mice

Tail tissue samples were heated in 50 µL digestion buffer (0.2 M Tris, 0.1 M (NH_4_)_2_SO_4_, 0.05 MgCl_2_, 1% beta-mercaptoethanol, 0.5% Triton-X 100, autoclaved Milli Q water) at 95 °C for 10 min. To digest the tail tissue and extract mouse genomic DNA Proteinase K (2 μg/uL, #AM2544, Invitrogen, Thermo Fisher Scientific, Waltham, MA, USA) was added and the samples were incubated overnight at 55 °C. The crude extracts were then heated for 10 min at 95 °C to inactivate the Proteinase K activity and centrifuged at 16,200× *g* to pellet insoluble material. Then, 0.5 µL of the crude DNA samples was used for the PCR. For genotyping of the mMCP-4 littermate mice the following primers were used: Forward 5′-CAA GGT CCA ACT AAC TCC CTT TGT GCT CC-3′, Reverse 5′-GGT GAT CTC CAG ATG GGC CAT GTA AGG GCG-3′, and Reverse Neo-cassette 5′-GGG CCA GCT CAT TCC TCC CAC TCA TGA TCT-3′, which yielded the expected 700 bp WT and 550 bp KO products. PCR was performed with the KAPA2G Robust HotStart PCR kit (#KK5517, Techtum Lab Ab, Stockholm, Sweden). 

### 2.5. Nested PCR Assay for Detection of Giardia DNA

Faecal samples 13 dpi or intestines 8 dpi were treated as described for the tail tissue to recover DNA. *Giardia* DNA was demonstrated using a nested PCR with the following primers targeting β-giardin: first round: Forward 5′-AAA TNA TGC CTG CTC GTC G-3′ and Reverse 5′-CAA ACC TTN TCC GCA AAC C-3′ and the second round: Forward 5′-CCC TTC ATC GGN GGT AAC TT-3′ and Reverse 5′-GTG GCC ACC CAN CCC GTG CC-3′, yielding a 530 bp product. PCR was performed with the KAPA2G Robust HotStart PCR kit (#KK5517, Techtum Lab Ab).

### 2.6. Sampling of Tissues and Morphological Staining

Pathological changes in the small intestine 8 dpi were assessed by light microscopy of formalin fixed and paraffin embedded tissue from mice in experiment #6. In brief, a ≈2–3 cm piece of the duodenum/jejunum was collected and fixed in 10% formalin for at least 24 h, embedded in paraffin, and sectioned on a microtome Microm HM355S (Thermo Fisher Scientific, Waltham, MA, USA). Five µm tissue sections were mounted on slides and stained with acidic toluidine blue (pH < 2.0) or with hematoxylin and eosin (H&E). The numbers of submucosal mast cells were counted in the whole tissue sample and the granulocytes were counted in ≥30 villi crypt units (VCU) per mouse intestine.

### 2.7. Determination of Intestinal Trypsin-Like, Chymotrypsin-Like, Neutrophil Elastase (NE), and Myeloperoxidase (MPO) Activity

To assess intestinal trypsin-like and chymotrypsin-like activity, small intestinal tissue from mice in experiment #6 was frozen in liquid nitrogen and homogenized with a mortar and pestle. The tissue powder was re-suspended in Hank’s balanced salt solution (HBSS) at 1 mL per 50 mg tissue and centrifuged at 15,000× *g* for 30 min at 4 °C. For trypsin-like activity 100 µL of the obtained supernatants (in triplicates) were incubated with 20 µL of 5 mM tryptase substrate (#S-2288, Chromogenix, Munich, Germany). For chymotrypsin-like activity 20 µL of the obtained supernatants (in triplicates) were incubated with 20 µL of 10 mM tryptase substrate (#L-1595, Bachem, Bubendorf, Switzerland). The optical density (OD) at 405 nm was determined at 0, 10, 30, and 120 min, and the difference over time was calculated. The enzyme activity is expressed as ΔOD.

To assess the intestinal neutrophil elastase (NE) activity, small intestinal tissue from mice in experiment #6 was frozen in liquid nitrogen and homogenized with a mortar and pestle. The tissue powder was re-suspended in Hank’s balanced salt solution (HBSS) at 1 mL per 50 mg tissue and centrifuged at 15,000× *g* for 30 min at 4 °C. 50 µL of the obtained supernatants were incubated with 15 µL of 10 μM elastase substrate aSuc-Ala-Ala-Pro-Val-AMC (#L-1770, Bachem) in 135 µL of the NE reaction buffer (150 mM NaCl, 100 mM Tris-HCl, 0.1% BSA, 0.05% Tween-20, MilliQ water, pH = 8.5). The optical density (OD) at 405 nm was determined at 0 min and after 60 min, and the difference was calculated. The enzyme activity is expressed as milli ΔOD per min per mg tissue.

To assess the activity of recombinant human neutrophil elastase (rhNE, R&D Systems, Minneapolis, MN, USA) and the potential effect of sGPs and ESPs, 0.026 μg rhNE were mixed with 15 μg sGPs or 4 µL ESPs in a 96-well microplate and incubated with 10 µL of 10 μM substrate in a final volume of 200 µL of NE reaction buffer. The absorbance was measured at 405 nm after 2 h of incubation at room temperature.

To assess the myeloperoxidase (MPO) activity, intestinal tissue frozen in liquid nitrogen was homogenized with a mortar and pestle and dissolved in 400 μL ice cold 1% hexadecyl trimethyl ammonium bromide in phosphate buffer (pH = 6.0) per 50 mg tissue. After snap freezing and thawing three times in liquid nitrogen, the homogenate was centrifuged 12000× *g* for 15 min at 4 °C. Then, 10 μL of the supernatant was added into 200 μL of the MPO substrate solution (50 mmol/L phosphate buffer, 0.4 mg/mL o-phenylenediamine substrate and 0.05% H_2_O_2_). The reaction was stopped after 20 min by adding of 50 μL 0.4 mol/L H_2_SO_4_, and absorbance was measured at 490 nm.

### 2.8. ELISA Assay for IL-6 Detection in Serum and Intestinal Tissues

Blood samples were collected at the experimental endpoint from mice in experimental infections #2 and #6 and allowed to clot. Serum was isolated by centrifugation 3000 rpm for 10 min at 4 °C. The small intestinal tissues from mice in experiments #3a and #6 were frozen in liquid nitrogen and homogenized with a mortar and pestle. The tissue powder was dissolved in RIPA buffer (150 mM NaCl, 1% Nonidet P-40, 0.5% Sodium deoxycholate, 0.1% SDS and 25 mM pH 7.4 of Tris) at 500 μL per 50 mg tissue and frozen/thawed three times. Supernatants were collected with centrifugation 10,000 rpm for 10 min at 4 °C. The concentration of IL-6 was determined in serum and intestinal samples, using a mouse IL-6 ELISA developmental kit (#900-T50, PeproTech Nordic, Stockholm, Sweden) according to the supplier’s protocol.

### 2.9. RNA Extraction and qPCR Detection of Chemokine and Cytokine Expressions in the Intestine

To extract total RNA, the small intestines from the mice in experimental infections #2, #3a, and #6 that were closest to the average of the weight curves (Figure 1a) were frozen in liquid nitrogen and homogenized with a mortar and pestle. The tissue powder was resuspended in TRIzol^®^ (Thermo Fisher Scientific) followed by RNA extraction according to the manufacturer’s instructions. DNase I treatment of extracted RNA was incorporated into the procedure to remove traces of genomic DNA. The quality of extracted total RNA was assessed by measuring the 260/280 and 260/230 ratios using a NanoDrop 1000 Spectrophotometer (Thermo Fisher Scientific). The relative expression of the intestinal expression of the following chemokines and cytokines were analyzed by quantitative real-time polymerase chain reaction (qPCR): CXCL-2, CCL-2, and IL-2, IL-4, IL-5, IL-6, IL-9, IL-10, IL-17a, IL-17c, IL-25, IL-33, TGF-β, and TNF-α. All qPCR primer pairs were designed in-house using the Primer-BLAST software at the National Center for Biotechnology Information (see Table 2 for the primer pairs). One μg of total RNA was reverse transcribed to cDNA using the RevertAid H Minus First Strand cDNA Synthesis Kit (Thermo Fisher Scientific) according to the manufacturer’s instructions. Maxima SYBR Green/ROX qPCR Master Mix (Thermo Fisher Scientific) was used for the qPCR and the expression of GAPDH was used for normalization according to guidelines of AB Applied Biosystems (Step One Plus Real Time PCR systems). All qPCR reactions were run in a Step One Plus Real Time PCR machine (Applied Biosystems, Thermo Fisher Scientific) using the following cycling conditions: activation of polymerase at 95 °C (15 s), annealing at 60 °C (30 s), and extension at 72 °C (30 s), followed by melting curve analysis as part of the default run settings. The fold change in gene expression between non-infected and *Giardia*-infected mMCP-4^+/+^ and mMCP-4^−/−^ mice was then calculated using the Livak-method (2^−ΔΔCT^) [43].

### 2.10. Statistical Analysis

The GraphPad Prism software (San Diego, CA, USA) was used for statistical analysis of the collected data. The non-parametric Mann–Whitney U test (not assuming normal distribution) was applied on the in vivo parameters and Student’s t-test were used elsewhere. The *p* values of <0.05 were considered as significant.

## 3. Results

### 3.1. Chymase-Deficient Mice Show Increased Weight Loss during Giardia Intestinalis Infections 

To address if the mast cell-specific protease chymase mMCP-4 would have an impact on the protection of the host during infection with *G. intestinalis*, we infected seven to 13 weeks old (referred to as young) and 18 to 36 weeks old (referred to as mature adult) congenic mMCP-4^+/+^, mMCP-4^+/−^ and chymase-deficient mMCP-4^−/−^ littermate mice in a series of completely blinded experiments (Table 1). For the young mice, weight data were pooled from three independent experiments, and for the adult mice, weight data from four independent experimental infections were pooled. Infected young female and male mice showed no significant changes in weight gain as compared with non-infected littermate controls (Appendix A). In contrast, infected mature adult mice significantly lost weight compared to non-infected littermate controls (Figure 1a). Infected mature adult female and male mice showed a similar trend in weight loss, but the changes were larger in the female mice during the first four days after infection (Figure 1b). The infected mature adult mMCP-4^−/−^ mice showed a significant weight loss already from day 2, whereas the mMCP-4^+/+^ mice showed significant weight loss from day 11, as compared with non-infected littermate mice (Figure 1a). *G. intestinalis* DNA was detected by nested PCR in feces collected 8 dpi and 13 dpi from both young and mature adult mice and in intestinal tissue from young mice at 8 dpi (Appendix A and data not shown). Since the effects of the *Giardia* infection on the growth performance was most evident in the mature adult mice, only data from this age group are presented in the following results.

### 3.2. Giardia-Infected Mice Show Increased Mast Cell and Granulocyte Counts in the Small Intestine but Decreased Neutrophil Elastase Activity

The morphological assessment of toluidine blue (TB)-stained formalin fixed and paraffin embedded intestinal sections from infected mMCP-4^+/+^ and mMCP-4^−/−^ mature adult mice (*n*^+/+^ = 8, *n*^−/−^ = 4, experiment #6) showed increased submucosal mast cell counts (Figure 2a) and increased mast cell activation (Figure 2b). The increased mast cell counts correlated with an increased trypsin-like activity in the infected mice (Figure 2c). The chymotrypsin-like activity was lower in the uninfected mMCP-4^−/−^ mice and the *G. intestinalis* infection caused a non-significant increase of the chymotrypsin-like activity (Figure 2d). However, the mast cell activities and mast cell counts were not significantly different between the infected mMCP-4^+/+^ and mMCP-4^−/−^ mice (Figure 2). 

Morphological assessment of hematoxylin eosin (H&E)-stained intestinal sections from infected mature adult mice (*n*^+/+^ = 7, *n*^−/−^ = 4, experiment #6) showed increased granulocyte counts at 8 dpi (Figure 3a). However, intestinal granulocyte counts were not significantly different between the mMCP-4^+/+^ and mMCP-4^−/−^ mice. Since H&E staining of the intestinal tissue may poorly resolve the ratio of neutrophils to eosinophils, we next assessed potential neutrophil-derived activities in the intestine by measuring neutrophil elastase (NE) activity and myeloperoxidase (MPO) activity. Strikingly, the NE activity decreased in infected mice (Figure 3b), while the MPO activity was unaffected (Figure 3c). The decreased activity of NE was unexpected and to further evaluate the NE enzymatic activity we measured the activity of recombinant human NE when challenged with *G. intestinalis* proteins. Soluble *Giardia* proteins (sGPs), but not *Giardia* Excretory Secretory proteins (ESPs), significantly inhibited the NE activity (Figure 3d). Neither the sGPs nor the ESPs had any intrinsic activity for the substrate L-1770 (Figure 3d).

### 3.3. Increased Expression Levels of IL-6 in Serum and Intestinal Tissue in Giardia-Infected Mice

IL-6 plays a significant role in the host control of *G. intestinalis* infections [29] and IL-6 levels can be regulated by the mast cell-specific chymase mMCP-4 [32]. Interestingly, the IL-6 serum levels were significantly increased in infected mice at 8 dpi (*n*^+/+^ = 5, *n*^−/−^ = 4, experiment #6) and 13 dpi (*n*^+/+^ = 8, *n*^−/−^ = 8, experiment #2) (Figure 4a), whereas no significant differences were recorded for the intestinal IL-6 levels (Figure 4b). The IL-6 levels were higher at 8 dpi than at 13 dpi. The lack of mMCP-4 in adult mice did not substantially affect the serum or the intestinal levels of IL-6 as compared with the mMCP-4^+/+^ littermate mice. However, the qPCR analysis of the intestinal expression of IL-6 and TNF-α in mice from experiments #2 (*n*^+/+^ = 3, *n*^−/−^ = 1), #3a (*n*^+/+^ = 1, *n*^−/−^ = 3) and #6 (*n*^+/+^ = 6, *n*^−/−^ = 4) showed significantly reduced up-regulation of these gene transcripts in adult mMCP-4^−/−^ infected mice as compared with the mMCP-4^+/+^ littermate mice (Figure 4c,d).

### 3.4. Giardia-Infected mMCP-4-Deficient Mice Show Reduced Expression of Alarmin and Chemokine Expression

To further delineate why the infected mMCP-4^−/−^ adult mice showed increased weight loss, we analyzed the transcriptional levels of IL-25 and the alarmin IL-33 as well as the chemokines CCL2 and CXCL2. Strikingly, the up-regulation of IL-25 expression seen in the mature adult mMCP-4^+/+^ was significantly reduced in the mMCP-4^−/−^ mice 8 dpi and 13 dpi (Figure 5a), whereas a minor down-regulation of the IL-33 expression was noted 8 dpi and 13 dpi in the infected mice (Figure 5b). The expression of CCL2 was up-regulated in infected mice as compared with the non-infected mice, with no major difference between the mMCP-4^−/−^ and mMCP-4^+/+^ mice (Figure 5c). Furthermore, the expression of CXCL2 was significantly reduced in mature adult mMCP-4^−/−^ mice as compared with the mMCP-4^+/+^ mice, both 8 dpi and 13 dpi (Figure 5d). 

### 3.5. Giardia-Infection in mMCP-4-Deficient Mice Affects the Th1-, Th2-, Th17-, and Treg-Induced Cytokine Profile 

To expand on the potential roles of mMCP-4 during infection with *G. intestinal* the expression levels of the major T cell-induced cytokine pathways in the small intestine at 8 dpi and 13 dpi were evaluated. The *Giardia*-infection induced changes in the Th1, Th2, Th17, and Treg transcriptional profiles in mature adult mice compared to non-infected littermate mice (Figure 6). In the mMCP-4^−/−^ mice, the transcriptional level of the Th1 cytokine IL-2 was significantly less up-regulated as compared to the mMCP-4^+/+^ mice (Figure 6a). Furthermore, the IL-4 and IL-5 expression levels were not significantly upregulated in the mMCP-4^−/−^ mice (Figure 6b), while the expression of the Th17 cytokines IL-17a and IL-17c were up regulated in the mMCP-4^+/+^ mice as compared to the mMCP-4^−/−^ mice (Figure 6c). The expression of the Treg cytokine TGF-β was not significantly different between mMCP-4^+/+^ and mMCP-4^−/−^ mice or non-infected littermate mice (Figure 6d). In contrast, we observed a significantly reduced expression level of IL-10 in the adult mMCP-4^−/−^ mice as compared to mMCP-4^+/+^ mice (Figure 6d).

## 4. Discussion

The chymase mMCP-4 is only significantly expressed by the connective tissue mast cells (CTMCs) (see reference [44] and Immgen.org), and the knockout of mMCP-4 did not affect the storage and expression of other mast cell proteases [45]. This stands in contrast to the knockout of mMCP-5, which affected the storage and activity of carboxypeptidase A3 [46] and vice versa, the knockout of CPA3 affected the storage and activity of mMCP-5 [47]. Furthermore, knocking out the enzyme Ndst-2, which replaces acetyl-groups with negatively charged sulphate groups on heparin, affects storage of histamine and the mast cell-specific proteases [48]. In addition, knocking out histamine synthesis by deletion of histidine decarboxylase, affects the storage of the mast cell specific proteases [49]. Importantly, we have previously shown that mast cell numbers were not significantly different in the evaluated tissues of mMCP-4^+/+^ and mMCP-4^−/−^ mice, naïve or challenged [50,51,52,53]. In contrast, the lack of chymase/mMCP-4 significantly reduced the infiltration of mast cells in the inflamed joint in a collagen-induced arthritis model [54], and in a model of bleomycin-induced lung inflammation, the mast cell numbers were increased in the mMCP-4^−/−^ mice [55]. In the mouse small intestine, the submucosal mast cell counts is normally very low (<1 mast cell per 100 villi) [56], and the low numbers of submucosal mast cells are confirmed by the low numbers of mast cell protease transcriptional reads ([44] and Lars Hellman, personal communication). During infections with *Giardia* and other intestinal parasites, submucosal mast cell counts increase ([23,24,25,57,58] and current study). However, in this *Giardia*-infection model, the number of submucosal intestinal mast cells were not affected by the lack of the mMCP-4 chymase.

In the intestine CTMCs reside in the submucosa/lamina propria closely associated with blood vessels. Therefore, a direct interaction with the non-invasive intestinal parasite *Giardia* and the mast cell-specific chymase may be difficult to envisage. However, during infections with helminth parasites, connective tissue mast cells migrate closer to the mucosal surfaces and may change their protease expression [57,59]. In addition, a recent study showed that activated urine bladder mast cells released granules containing chymase/mMCP-4, inducing apoptosis and shedding of the epithelial umbrella cells, to remove the infection with uropathogenic *E. coli* [60]. Furthermore, in the intestine mMCP-4 contributes to homeostatic epithelial cell migration and barrier function [34]. Thus, chymase mMCP-4 may act distantly from the CTMCs and could potentially affect several intestinal mucosal mechanisms. 

A role for mast cells in controlling the *G. intestinalis* infection has been demonstrated in vivo in several different animal models [19,20]. Earlier studies in mice used animals less than 10 weeks of age that were treated with antibiotics before the challenge infection [7,8,22,35,61,62]. To investigate the potential role of the mast cell-specific chymase mMCP-4, we established a gavage-infection model using congenic littermate mice. These mice were not treated with antibiotics in order to retain the intestinal bacteria flora before the experimental infection. Initially both mature adult and young (≈10 weeks old) congenic mMCP-4^+/+^, mMCP-4^+/−^ and mMCP-4^−/−^ littermate mice from our inhouse breeding colony were infected with *G. intestinalis* to address potential age-related differences. The infection resulted in significant weight loss in the mature adult mice at the experimental endpoint, although with an earlier weight loss in the mMCP-4^−/−^ mice as compared with the mMCP-4^+/+^ mice. The more rapid weight loss observed in the mMCP-4-deficient mice could depend on several chymase-dependent issues. For example, chymase has been shown to degrade the neuropeptide vasoactive intestinal polypeptide (VIP) [63] and VIP regulates intestinal motility [64]. We may speculate that the lack of chymase potentially leads to higher levels of VIP, an increase in intestinal motility, and earlier weight loss in the infected mMCP-4^−/−^ mice. We also noticed a faster weight decrease in adult female mice (Figure 1b). Earlier studies have shown that the sex and age of mice affect the mucus layer, the microbiota, and the intestinal immune system [40]. Old mice have a thinner Muc2 layer in the intestine, and the microbiota can get in contact with the intestinal epithelial cells [40]. *Giardia* prefers to bind to areas in the intestine without a mucus layer [65,66], and a reduced mucus layer can explain why the old mice are more sensitive to *Giardia* infections. The composition of the microbiota and the intestinal immune responses are different in old female and male mice [40], and this can potentially explain the differences in weight loss during *Giardia* infections. Another factor that is different in old female mice compared to males is the expression of tight and adherence junction proteins in the intestine [40]. *Giardia* parasites degrade tight junction proteins and induce a leaky gut [16,65] and old female mice express less tight junction proteins, potentially resulting in an increased leakiness. Thus, there are several potential causes to the increased weight loss in adult mice, but more experiments are needed in order to determine the causes. 

*G. intestinalis*-infected mMCP-4^+/+^ and mMCP-4^−/−^ mice showed a similar increase in submucosal mast cell counts, degranulation status and increased trypsin-like activity. However, we observed significantly reduced intestinal gene expression levels of alarmins, and cytokines in the *G. intestinalis*-infected mature adult mMCP-4^−/−^ mice. Lack of mMCP-4 caused significantly reduced up-regulation of the pro-inflammatory cytokines IL-6 and TNF-α in the *Giardia* infected mature adult mice although these differences did not reach significance at the protein level in blood or intestinal tissue. Both IL-6 and TNF-α can contribute to the elimination of *Giardia* [31], and IL-6 can modulate B cell maturation and induce antibody class switching to IgA during infection with *Giardia* [67]. 

IL-25 is exclusively expressed by the rare Tuft cells (brush cells) scattered throughout the small intestine. IL-25 and IL-33 secreted from epithelial cells and tissue dwelling sentinel cells has been reported to be essential for the expansion of innate lymphoid cells type 2 that initiates intestinal immune responses against helminths [68,69]. The effects of a *Giardia*-infection on the expression levels of IL-25 (IL-17E) and the alarmin IL-33 have, to our knowledge, never been studied. Interestingly, the intestinal expression of IL-25 at 8 dpi was significantly up-regulated in a mMCP-4-dependent manner in mature adult mice, suggesting that chymase may affect the expression of IL-25 from the Tuft cells. Alternatively, the lack of mMCP-4 may result in less numbers of intestinal Tuft cells, since epithelial cell migration is reduced in the mMCP-4^−/−^ mice [34]. We and others have shown that mMCP-4 can activate the pro-cytokine IL-33 and that human and mouse chymase then degrades active IL-33, which is likely to reduce excessive inflammation in the stressed tissue [70,71,72]. The observed increased IL-25 expression and determination of intestinal protein levels of IL-25 and IL-33 during infection with *G. intestinalis* warrants further investigation.

We also found that the presence or absence of mMCP-4 affect the RNA expression levels of IL-2, IL-4, IL-5, IL-17a, and IL-17c as well as IL-10 in the mature adult mice, suggesting that mMCP-4 could affect the intestinal recruitment of the cells responsible for the Th1, Th2, Th17, and Treg cytokine expression profiles, or possibly influence their transcriptional (re)-activation in the intestinal tissue. Although the Th1 cytokine IFN-γ and the Th2 cytokines IL-4 and IL-5 have been shown to be elevated in *Giardia*-infection models [31,73,74], both IFN-γ-deficient and IL-4-deficient mice can still eliminate *Giardia* [75], suggesting that the Th1 and Th2 responses are not essential for clearance. The IL-17 expression was up-regulated in cattle and mice infected with *Giardia* [73,76,77] and the IL-17a receptor-knockout mice failed to clear a *Giardia*-infection [77], suggesting that Th17 cells and neutrophil responses are important for elimination of the infection. In the *Giardia*-infected mature adult mMCP-4^−/−^ mice we observed a significantly reduced up-regulation of IL-5 at 8 dpi and 13 dpi as well as of IL-17a and IL-17c at 13 dpi. Despite that, the mMCP-4^−/−^ mice showed a similar intestinal granulocyte recruitment as the mMCP-4^+/+^ mice. 

To further assess the increased granulocyte counts, we analyzed the intestinal activity of neutrophil elastase (NE), a serine proteinase stored in resting neutrophil granules, which may cleave extracellular matrix components and cell surface molecules during tissue injury and inflammation [78]. The inhibition of NE prevents cathelicidin activation and impairs clearance of bacteria from wounds [79]. In mice infected with *Leishmania major*, it was reported that NE could reduce parasite burden and contribute to the killing via macrophage activation [80,81]. In addition, the NE activity increased significantly in *Thichinella spiralis*-infected mice [58]. Unexpectedly, the infected mMCP-4^+/+^ and mMCP-4^−/−^ mice showed a reduced NE activity as compared to the non-infected mice. This finding suggests that neutrophils either leave the intestine during a *Giardia* infection or possibly that *Giardia* can inhibit the NE activity. To test the latter, we challenged recombinant human NE with the addition of soluble *Giardia* proteins (sGPs), which significantly inhibited the NE activity. The inhibition was not complete, suggesting a competitive inhibition by sGPs, possibly by blocking the catalytic active site in NE. This observation suggests that *Giardia* produces and secretes NE inhibitors to counteract the potentially harmful activities of intestinal neutrophils, a finding that warrants further investigation.

In summary, the *Giardia*-infection caused a mMCP-4-dependent up-regulation of several cytokines in mature adult mice and the qPCR results suggest that mast cell chymase may promote and regulate the intestinal cytokine responses toward *G. intestinalis.* Thus, the mMCP4 knockout mouse strain provides an interesting model for studies of the ensuing intestinal immune responses during protozoan parasitic infections. However, this observation needs further investigations and the cell types responsible for the difference in the intestinal cytokine profiles between the mMCP-4^+/+^ and the mMCP-4^−/−^ mice needs to be defined.

## Figures and Tables

**Figure 1 cells-09-00925-f001:**
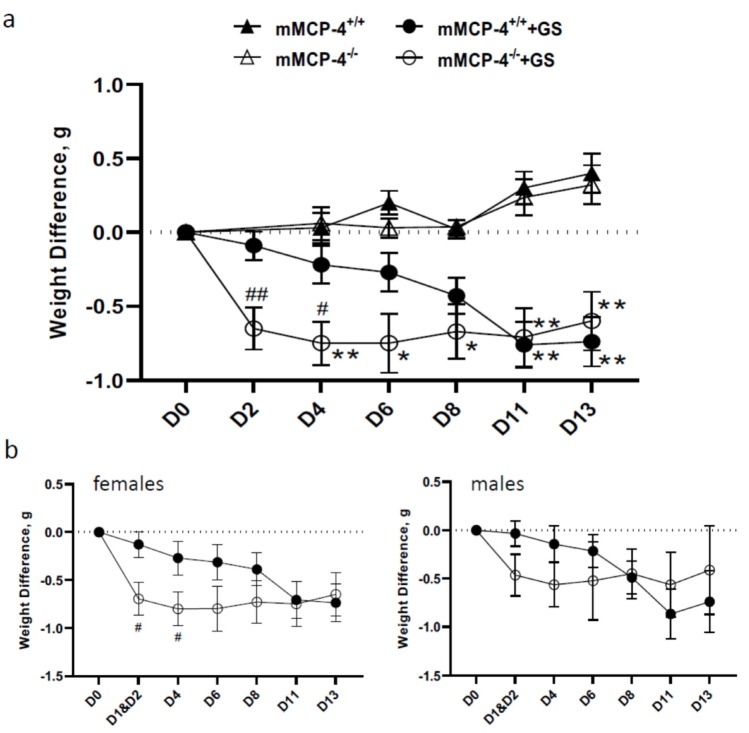
Weight loss in mMCP-4^−/−^ mice infected with *Giardia intestinalis*. Mature adult (18–36 weeks) congenic mMCP-4^+/+^, mMCP-4^−/−^ C57Bl/6 littermate mice without antibiotic treatment were infected by gavage with 10^6^
*G. intestinalis* GS trophozoites. Weight data were collected every second or third day before mice were euthanized on day 8, 13, or 22. Weight changes in gram were normalized to initial weight at day 0. (**a**) Pooled weight data from all mice in four independent infection experiments, i.e., #1, 2, 3a, and 6, are shown (see Table 1). Note that mMCP-4^+/+^ (*n* = 27) and mMCP-4^−/−^ (*n* = 24) infected mice showed significant weight loss as compared with the age matched mMCP-4^+/+^ (*n* = 3) and mMCP-4^−/−^ (*n* = 3) noninfected littermate mice. (**b**) Weight changes for female (mMCP-4^+/+^ (*n* = 16) and mMCP-4^−/−^ (*n* = 19)) and male (mMCP-4^+/+^ (*n* = 11) and mMCP-4^−/−^ (*n* = 5)) groups. *, ** represent *p* < 0.05 or 0.01 vs. non-infected mice, #, ## represent *p* < 0.05 or 0.01 vs. mMCP-4^+/+^ infected mice.

**Figure 2 cells-09-00925-f002:**
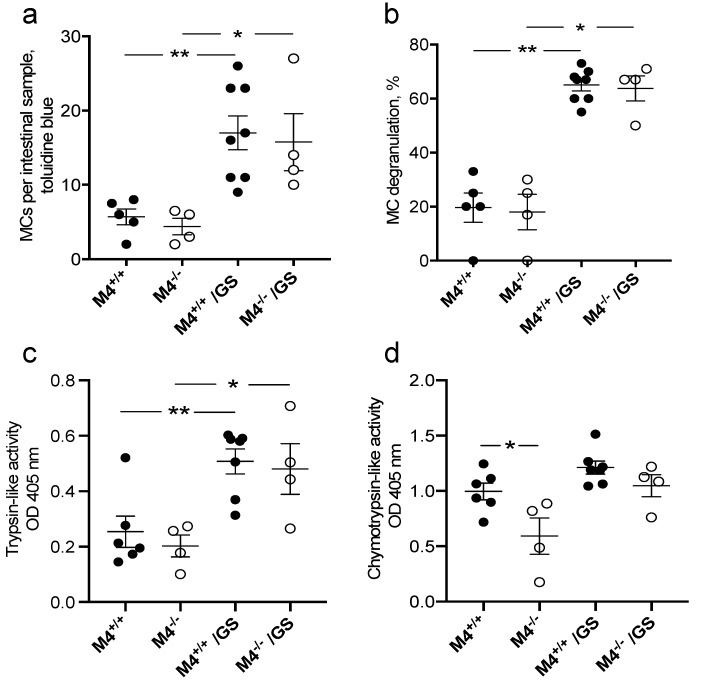
Increased mast cell counts, degranulation and trypsin-like activity in the small intestine during infection with *Giardia intestinalis*. Small intestinal sections from non-infected and >18-week-old mice 8 dpi were stained with toluidine blue (TB), and (**a**) submucosal mast cells were counted and (**b**) degranulation status of the mast cells were scored. (**c**) Trypsin-like and (**d**) chymotrypsin-like activity in intestinal tissue of 8 dpi mMCP-4^+/+^ and mMCP-4^−/−^ littermate mice and in non-infected littermate mice. * and ** represents *p* < 0.05 or 0.01 for infected mice vs. noninfected mice.

**Figure 3 cells-09-00925-f003:**
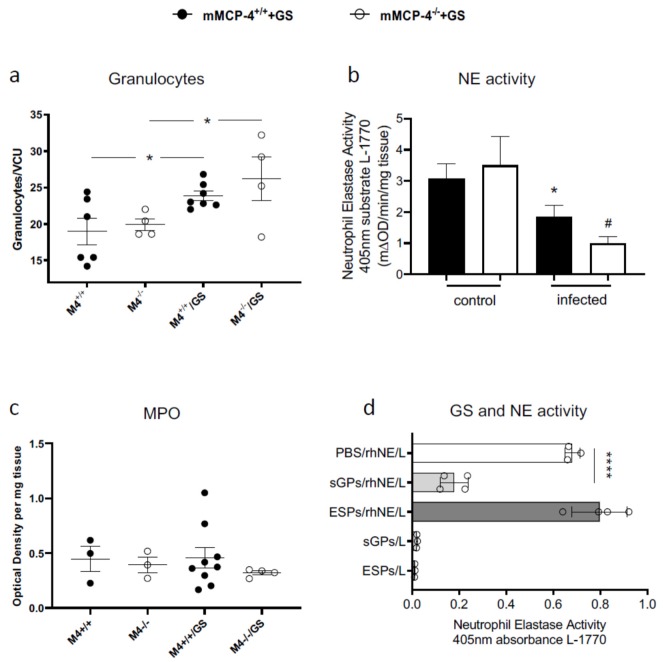
Increased granulocyte counts and reduced neutrophil elastase (NE) activity in the small intestine during infection with *Giardia intestinalis*. (**a**) Granulocytes in the villi and crypt area were counted on H&E-stained small intestinal sections from non-infected and >18-week-old mice 8 dpi. Intestinal NE activity (**b**) and MPO activity (**c**) in 8 dpi mMCP-4^+/+^ and mMCP-4^−/−^ littermate mice and in non-infected littermate mice. (**d**) The activity of recombinant human NE was challenged with sGPs or ESPs. VCU, villi crypt unit. sGPs, soluble *Giardia* proteins. ESP, *Giardia* excretory secretory proteins. L, the NE substrate L-1770. *, # represent *p* < 0.05 for infected vs. non-infected mice and **** represent *p* < 0.0001 for sGPs/rhNE vs. PBS/rhNE.

**Figure 4 cells-09-00925-f004:**
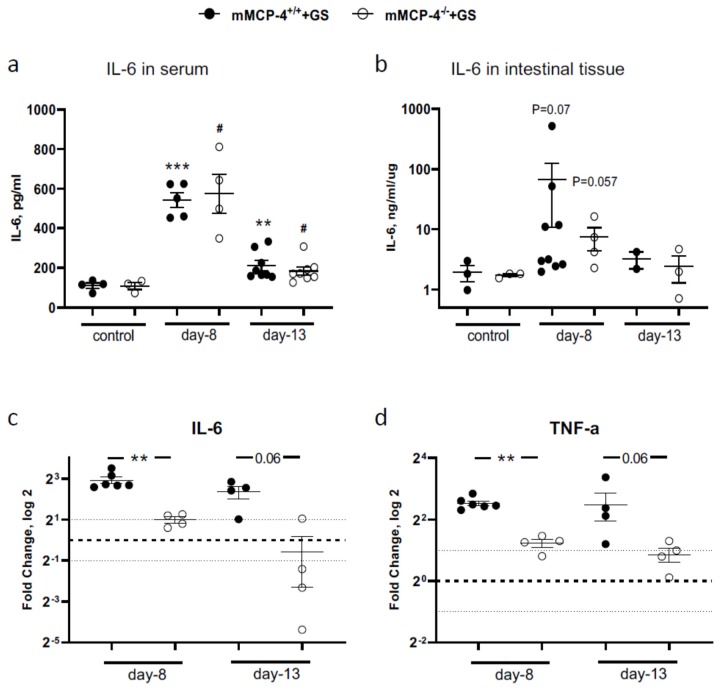
Pro-inflammatory cytokine expression levels in mMCP-4^+/+^ and mMCP-4^−/−^ mice infected with *Giardia intestinalis*. IL-6 protein levels were measured by ELISA in >18-week-old mice in (**a**) serum and (**b**) intestinal tissue homogenates from experiments #2 or #3a, and #6. Intestinal mRNA expression levels (qPCR) of IL-6 (**c**) and TNF-α (**d**) in *Giardia*-infected >18 weeks old mice from experiments #2, #3a and #6. Fold changes were calculated after normalization to GAPDH. The dotted line is at 1 and indicates no fold change. In a) **, *** indicate *p* < 0.01 or 0.001 vs. non-infected mMCP-4^+/+^ mice and # indicates *p* < 0.05 vs. non-infected mMCP-4-/-mice. In (c) and (d), ** indicates *p* < 0.01 for mMCP-4^+/+^ vs. mMCP-4^−/−^ mice.

**Figure 5 cells-09-00925-f005:**
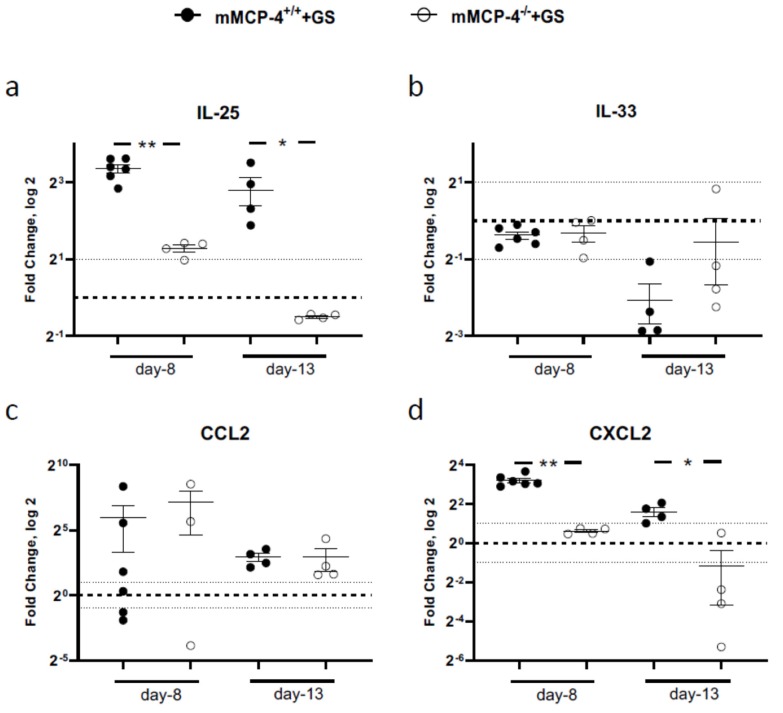
Alarmin and chemokine expression levels in mMCP-4^+/+^ and mMCP-4^−/−^ mice infected with *Giardia intestinalis*. Intestinal expression (qPCR) 8 dpi and 13 dpi of IL-25 (**a**) and the alarmin IL-33 (**b**) as well as the chemokines CCL2 (**c**) and CXCL2 (**d**) in *Giardia*-infected mature adult mice from experiments #2, #3a, and #6. Fold changes were calculated after normalization to GAPDH. The dotted line is at 1 and indicates no fold change. In (**a**) and (**d**), *, ** indicate *p* < 0.05 or 0.01 for mMCP-4^+/+^ vs. mMCP-4^−/−^ mice.

**Figure 6 cells-09-00925-f006:**
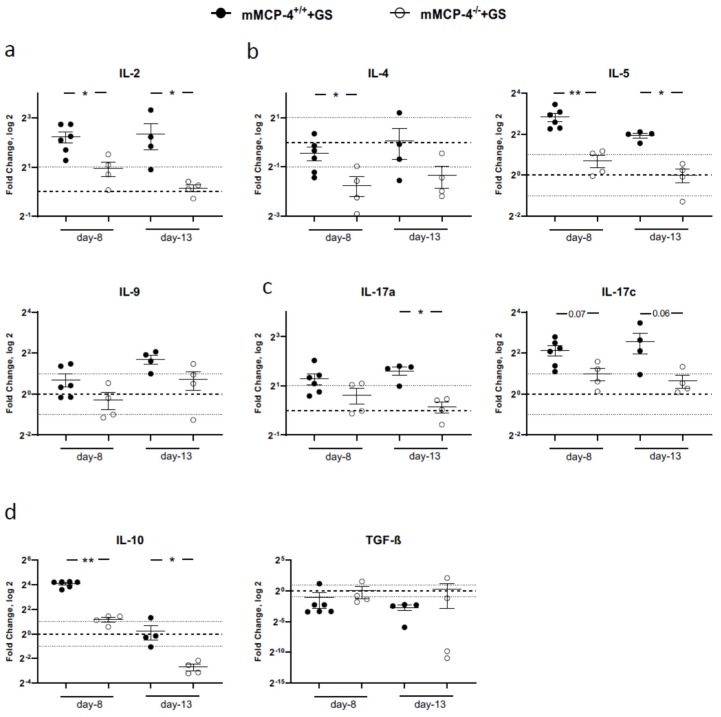
Th1-, Th2-, Th17-, and Treg-derived cytokine expression levels in mMCP-4^+/+^ and mMCP-4^−/−^ mice infected with *Giardia intestinalis*. qPCR analysis 8 dpi and 13 dpi of the expression levels of (**a**) IL-2, (**b**) IL-4, IL-5 and IL-9, (**c**) IL-17a and IL-17c, and (**d**) IL-10 and TGF-b, in *Giardia*-infected mature adult mice from experiments #2, #3a, and #6. Fold changes were calculated after normalization to the expression of GAPDH. The dotted line is at 1 and indicates no fold change. *, ** indicate *p* < 0.05 or 0.01 for mMCP-4^+/+^ vs. mMCP-4^−/−^ mice.

**Table 1 cells-09-00925-t001:** Summary of gavage infections with trophozoites (≈10^6^) of the *Giardia intestinalis* GS isolate in congenic mMCP-4^+/+^ and mMCP-4^−/−^ C57Bl/6 littermate mice without antibiotic treatment.

# Exp.	Gender	Number of Mice	Age (weeks)	Endpoint Days
#1	F	^+/+^*n* = 7, ^+/−^ *n* = 9, ^−/−^ *n* = 5	23–36	22
#2	F	^+/+^*n* = 5, ^+/−^ *n* = 6, ^−/−^ *n* = 9	22–29	13
	M	^+/+^*n* = 5, ^+/−^ *n* = 6, ^−/−^ *n* = 3		
#3a	F	^+/+^*n* = 1, ^+/−^ *n* = 2, ^−/−^ *n* = 2	18–20	13
	M	^+/+^*n* = 0, ^+/−^ *n* = 4, ^−/−^ *n* = 1		
#6	F	^+/+^*n* = 3, ^+/−^ *n* = 6, ^−/−^ *n* = 3	24	8
	M	^+/+^*n* = 6, ^+/−^ *n* = 5, ^−/−^ *n* = 1		
#3b	F	^+/+^*n* = 1, ^+/−^ *n* = 5, ^−/−^ *n* = 1	7–13	13
	M	^+/+^*n* = 1, ^+/−^ *n* = 5, ^−/−^ *n* = 2		
#4	F	^+/+^*n* = 1, ^+/−^ *n* = 3, ^−/−^ *n* = 3	10	13
	M	^+/+^*n* = 1, ^+/−^ *n* = 4, ^−/−^ *n* = 0		
#5	F	^+/+^*n* = 0, ^+/−^ *n* = 2, ^−/−^ *n* = 1	10	8
	M	^+/+^*n* = 2, ^+/−^ *n* = 3, ^−/−^ *n* = 2		

**Table 2 cells-09-00925-t002:** Forward and reverse primers used for qPCR.

Gene	Direction	Sequence
GAPDH	forward	CAAGCTCATTTCCTGGTATGACAAT
	reverse	CTCTCTTGCTCAGTGTCCTTGC
IL-25	forward	GCTCCAGTCAGCCTCTCTC
	reverse	CTGCTCACCAGTCACAGGT
IL-33	forward	TCTGCCCCTTCTTTGGTT
	reverse	GGGAGTAGGAGAGCCGTTAC
IL-6	forward	TGGGACTGATGCTGGTGAC
	reverse	CACAACTCTTTTCTCATTTCCACG
TNF-ALFA (BOTH VARIANTS)	forward	ACGGCATGGATCTCAAAG
	reverse	TGGGAGTAGACAAGGTACAACC
CCL-2	forward	CACTCACCTGCTGCTACTCATTC
	reverse	GGTGCTGAAGACCTTAGGGC
CXCL-2	forward	ATACTGAACAAAGGCAAGGCTAACTG
	reverse	CTCAGACAGCGAGGCACATC
IL-2	forward	TGTAAAACTAAAGGGCTCTGACA
	reverse	AGAAAGTCCACCACAGTTGCT
IL-12A (BOTH VARIANTS)	forward	GTCAATCACGCTACCTCCTCT
	reverse	CGGGACTGGCTAAGACAC
IL-4 (VARIANT 1)	forward	GCAACGAAGAACACCACAGAG
	reverse	GAAGCACCTTGGAAGCCCTA
IL-5	forward	GAAATACATTGACCGCCAAAAAGT
	reverse	GCCTCAGCCTTCCATTGC
IL-9	forward	CCTTGCCTCTGTTTTGCTCT
	reverse	ATCATCAGTTGGGACGGAGAG
IL-17A	forward	TCAGACTACCTCAACCGTTCC
	reverse	CTATCAGGGTCTTCATTGCG
IL-17C	forward	GAGATATCGCATCGACACAGA
	reverse	CATCCACGACACAAGCATT
TGFβ-1	forward	CTGCTGACCCCCACTGATAC
	reverse	AAAGCCCTGTATTCCGTCTCC
IL-10	forward	CCTGGTAGAAGTGATGCCCC
	reverse	ATTCAAATGCTCCTTGATTTCTGG

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
