# Peer review of "The Chymase Mouse Mast Cell Protease-4 Regulates Intestinal Cytokine Expression in Mature Adult Mice Infected with Giardia intestinalis"

_cells, 2020, doi:10.3390/cells9040925_

Round 1

Reviewer 1 Report

This study investigates the potential role of the chymase mouse mast cell protease-4 (mMCP-4) during experimental infections with Giardia infection in mature adult mice. Using mMCP-4+/+ and mMCP-4-/- littermate mice infected with G. intestinalis, the authors demonstrated that the lack of mMCP-4 caused more rapid significant weight loss in comparison with mMCP-4+/+ mice, and significantly less intestinal transcriptional upregulation of IL-6, TNF-a, IL-25, CXCL2, IL-2, IL-4, IL-5, and IL-10. Both genotypes of mice presented increased granulocyte counts in jejunum without increased myeloperoxidase and neutrophil elastase activity. To better understand this result, in vitro assay shows that soluble Giardia proteins inhibited neutrophil elastase. Altogether, the manuscript is well written and adds to our knowledge the potent immunoregulatory roles of mMCP-4 in G. intestinalis infection. However, there are points in the text that require clarification, as follows:

Considering that inactivation of genes for some mast cell enzymes resulted in gross mast cell morphology and reduction in other stored proteases (Stevens et al., FASEB J. 10:17772, 1996; Forsberg et al., Nature, 400:773–776, 1999; Ohtsu et al., FEBS Lett., 502:53–56, 2001), does the lack of mMCP-4 cause any alteration in the mast cell population or storage of another protease? It is important to clarify this aspect in the text.   In this model of infection, is there an increase in the mast cell population? In the methods, section 2.6, it is described that intestine sections were stained with toluidine blue, a classic stain for detection of mast cells, but no description about this analysis appears in results. I believe that is crucial to demonstrate the presence of mast cells in the intestine to correlate with Giardia infection.

Minor point:

Check the legend of figure 2 for double “c)”. Include “d)” for correct description of panel d.

Author Response

Answers to reviewer comments

We have carefully checked the language, grammar and spelling throughout the manuscript.

Changes made in the manuscript is highlighted in red text.

We have included a new figure.

Reviewer 1: This study investigates the potential role of the chymase mouse mast cell protease-4 (mMCP-4) during experimental infections with Giardia infection in mature adult mice. Using mMCP-4+/+ and mMCP-4-/- littermate mice infected with G. intestinalis, the authors demonstrated that the lack of mMCP-4 caused more rapid significant weight loss in comparison with mMCP-4+/+ mice, and significantly less intestinal transcriptional upregulation of IL-6, TNF-a, IL-25, CXCL2, IL-2, IL-4, IL-5, and IL-10. Both genotypes of mice presented increased granulocyte counts in jejunum without increased myeloperoxidase and neutrophil elastase activity. To better understand this result, in vitro assay shows that soluble Giardia proteins inhibited neutrophil elastase. Altogether, the manuscript is well written and adds to our knowledge the potent immunoregulatory roles of mMCP-4 in G. intestinalis infection. However, there are points in the text that require clarification, as follows:

Considering that inactivation of genes for some mast cell enzymes resulted in gross mast cell morphology and reduction in other stored proteases (Stevens et al., FASEB J. 10:17772, 1996; Forsberg et al., Nature, 400:773–776, 1999; Ohtsu et al., FEBS Lett., 502:53–56, 2001), does the lack of mMCP-4 cause any alteration in the mast cell population or storage of another protease? It is important to clarify this aspect in the text.  

  • We thank the reviewer for pointing-out this important issue. In the first publication of the mMCP-4 knockout mouse strain (Tchougounova, Pejler, Abrink, J Exp Med. 2003 198(3):423-431) we analyzed the gross mast cell morphology in peritoneal lavage cells. The mMCP-4-deficient mast cells had similar gross morphology as the mMCP-4-competent mast cells and were found to degranulate equally to wild type mast cells, when activated by calcium ionophore A23187 or by adding an anti-IgE Furthermore, the lack of mMCP-4 in peritoneal mast cells as well as in ear tissue did not affect the overall content and activity of the other granule-stored mast cell proteases, i.e. mMCP-5, mMCP-6 and CPA3, when analyzed with Western blot and with chromogenic substrates for the respective proteases (S-2288 for trypsin-like proteases and M-2245 for CPA3). See figure 2 and 3 as well as Table I and II in the paper, Tchougounova et al. J Exp Med. 2003 198(3):423-43. Furthermore, we have in several additional studies carefully evaluated the number of mast cells in mMCP-4+/+ and mMCP-4-/- littermate mice in different tissues, before and after challenge in the different disease models. For example, mast cell numbers were not significantly different in the evaluated tissues of mMCP-4+/+ and mMCP-4-/- mice, naïve or challenged (Tchougounova et al. JBC 2005; Waern et al. J Immunol. 2009; Scandiuzzi et al. J Immunol. 2010; Lin et al. JBC 2011). In contrast, the lack of chymase/mMCP-4 reduced joint mast cell numbers in a collagen-induced arthritis model (Magnusson et al. FASEB J. 2009) and in a model of bleomycin-induced lung inflammation the mast cell numbers were increased in the mMCP-4-/- mice (Reber et al. J Immunol. 2014).

To clarify the issue around mast cell protease content we have added the following text with additional references into the first paragraph of the discussion.

The knockout of mMCP-4 do not affect the storage and expression of other mast cell proteases (Tchougounova et al. 2003). This stands in contrast to the knockout of mMCP-5, which affected the storage and activity of carboxypeptidase A3 (Stevens et al. 1996), and vice versa, the knockout of CPA3 affected the storage and activity of mMCP-5 (Feyerabend et al. 2005; Schneider et al. 2007). Furthermore, knocking out the enzyme Ndst-2, which replaces acetyl-groups with negatively charged sulphate groups on heparin, affects storage of histamine and the mast cell-specific proteases (Humphries et al. 1999; Forsberg et al. 1999). In addition, knocking out histidine decarboxylase, that synthesizes histamine, affects the storage of the mast cell specific proteases (Ohtsu et al. 2001). Importantly, we have previously shown that mast cell numbers were not significantly different in the evaluated tissues of mMCP-4+/+ and mMCP-4-/- mice, naïve or challenged (Tchougounova et al. 2005; Waern et al. 2009; Scandiuzzi et al. 2010; Lin et al. 2011). In contrast, the lack of chymase/mMCP-4 significantly reduced the infiltration of mast cells in the inflamed joint in a collagen-induced arthritis model (Magnusson et al. 2009), and in a model of bleomycin-induced lung inflammation the mast cell numbers were increased in the mMCP-4-/- mice (Reber et al. 2014). However, in this Giardia-infection model the number of submucosal intestinal mast cells were not affected by the lack of chymase.”

In this model of infection, is there an increase in the mast cell population? In the methods, section 2.6, it is described that intestine sections were stained with toluidine blue, a classic stain for detection of mast cells, but no description about this analysis appears in results. I believe that is crucial to demonstrate the presence of mast cells in the intestine to correlate with Giardia infection.

  • We thank the reviewer for mentioning this important issue. As pointed out by the reviewer toluidine blue (TB) is a valid stain for submucosal mast cells, e. connective tissue mast cells (CTMCs), in most tissues. Successful staining of CTMCs depends on the pH of the TB staining, and usually requires a pH set below 2 (pH <2). Expression of the chymase mMCP-4 is highly restricted to CTMCs (see Immgen http://www.immgen.org, where the tool Gene SkyLine (containing RNA-seq data and microarray data) shows detailed analysis of expression levels). Mouse mucosal mast cells MMCs do not express mMCP-4, instead they express mMCP-1 and mMCP-2. During infections with helminths submucosal intestinal mast cells may indeed express mMCP-4 but as the mast cells migrate out into the mucosa and become intraepithelial MMCs they shift to express mMCP-1 and mMCP-2 (Friend et al. J Cell Biol. 1996 135(1). 279-290; Friend et al. J Immunol. 1998 160(11):5537-45; Vogel et al. Veterinary Pathology 2018 55(1):76-97). Importantly, staining for submucosal and mucosal mast cells with hematoxylin and eosin (HE), Periodic Acid-Schiff (PAS), Alcian blue and Luna in formalin-fixed paraffin embedded (FFPE) intestinal tissues will, however, turn out negative (Vogel et al. Veterinary Pathology 2018 55(1):76-97). We have now prepared more sections from the FFPE intestinal tissues and stained with Luna, HE and TB (pH <2). And as described by Vogel et al., we also fail to differentially stain the mast cells in the small intestinal tissue using Luna and HE staining protocols. But since mMCP-4 is restricted to expression in CTMCs we therefore focused on the characterization of the submucosal mast cells. In TB stained (pH <2) mature adult intestinal tissues from non-infected mice and day 8 infected mice the counting of CTMCs in the intestinal lamina propria close to the smooth muscle layer are shown in new figure 2. This shows an increased level of submucosal mast cells in the intestinal epithelium during Giardia infections. This data and comments are now found in the paper.

Minor point:

Check the legend of figure 2 for double “c)”. Include “d)” for correct description of panel d. 

  • Thank you, we have now corrected the figure legend.

The old figure 2 is now new figure 3.

The old figure 3 is now new figure 4.

The old figure 4 is now new figure 5.

The old figure 5 is now new figure 6.

Reviewer 2 Report

This is an interesting study attempting to mechanistically link mMCP-4 to Giardia instetinalis (GI) Infection in the mouse model.

Some concerns are however stated below.

1- (Figure 1 b) The authors do not provide an explanation as to why GI infected female mMCP-4 KO mice loose more weight than their male congeners.

2- (Figure 3 d and e; Figure 5, Figure 6)  Increased expression of most of the cytokines studied are not experimentally correlated to their respective protein levels. It would be interesting to show these protein levels.

3- The authors do not provide information nor experimental evidences to explain how mMCP-4 modulates expression of these cytokines. 

4- It would also be recommended to the authors to make sure that total mastocyte counts in peritoneal lavages are not significantly different in WT versus mMCP-4 -/- mice, naive or GI-infected. 

Author Response

Answers to the reviewer comments.

Changes made in the manuscript is highlighted in red text.

We have included a new figure 2.

The old figure 2 is now new figure 3.

The old figure 3 is now new figure 4.

The old figure 4 is now new figure 5.

The old figure 5 is now new figure 6.

Reviewer 2: This is an interesting study attempting to mechanistically link mMCP-4 to Giardia instetinalis (GI) Infection in the mouse model.

  • We thank the reviewer for the positive comment.

Some concerns are however stated below.

1- (Figure 1 b) The authors do not provide an explanation as to why GI infected female mMCP-4 KO mice loose more weight than their male congeners.

  • We don’t think that Figure 1b gives room for the interpretation made by the reviewer. Figure 1b shows that there is a similar trend in earlier weight loss for the infected mMCP-4 KO mice, both females (significant) and males (not significant) as compared with WT mice. Both WT and mMCP-4 KO mice in fact have lost similar in weight by day 8 and day 13 as compared with the uninfected animals.

2- (Figure 3 d and e; Figure 5, Figure 6)  Increased expression of most of the cytokines studied are not experimentally correlated to their respective protein levels. It would be interesting to show these protein levels.

  • We thank the reviewer for pointing this out. However, unfortunately the limited intestinal material derived from each individual mouse, is not enough for the suggested in-depth analysis of all cytokines with changed expression at the protein level. We have used the jejunal part of the small intestine and at end point we cut this into 2-3 cm pieces, the pieces going into RNA-later or formalin, or being snap-frozen. One small intestinal piece was used for RNA extraction and qPCR, one piece was used for studies of intestinal morphology and to do different histo-stains, and two pieces were used for protein-extracts to evaluate the activities of different enzymes (neutrophil elastase, myeloperoxidase and mast cell protease activity) with substrates or for cytokine level determination by ELISA. However, we have changed the title of the manuscript to better reflect the data presented here.

3- The authors do not provide information nor experimental evidences to explain how mMCP-4 modulates expression of these cytokines. 

  • This is an important question and today we do not have any experimental evidence which explains this regulation. We feel that such a characterization is beyond the scope of the current study, which is describing the effect for the first time. A follow-up will require many more mouse experiments as the major clinical difference between the WT and mMCP-4 KO mice occurs before day 8. To resolve these issues immunohistochemical (IHC) stainings with several specific reagents should be performed. Unfortunately, we do not have the material from these infections, nor established methods to perform this type of studies today. However, this is definitively something we would like to continue with and future studies using this experimental system at earlier time-points can show how mMCP-4 affect the expression of cytokines.

4- It would also be recommended to the authors to make sure that total mastocyte counts in peritoneal lavages are not significantly different in WT versus mMCP-4 -/- mice, naive or GI-infected. 

  • We thank the reviewer for pointing to this important issue. However, the Giardia infection is not invasive and we have therefore concentrated our characterization of the mast cells situated in the intestinal tissue (and not mesenteric mast cells). We have added new data concerning the level of mast cells in the small intestine during the infections. Expression of the chymase mMCP-4 is highly restricted to CTMCs (see Immgen http://www.immgen.org, where the tool Gene SkyLine (containing RNA-seq data and microarray data) shows detailed analysis of expression levels). Mouse mucosal mast cells MMCs do not express mMCP-4, instead they express mMCP-1 and mMCP-2. During infections with helminths submucosal intestinal mast cells may indeed express mMCP-4 but as the mast cells migrate out into the mucosa and become intraepithelial MMCs they shift to express mMCP-1 and mMCP-2 (Friend et al. J Cell Biol. 1996 135(1). 279-290; Friend et al. J Immunol. 1998 160(11):5537-45; Vogel et al. Veterinary Pathology 2018 55(1):76-97). Importantly, staining for submucosal and mucosal mast cells with hematoxylin and eosin (HE), Periodic Acid-Schiff (PAS), Alcian blue and Luna in formalin-fixed paraffin embedded (FFPE) intestinal tissues will, however, turn out negative (Vogel et al. Veterinary Pathology 2018 55(1):76-97). We have now prepared more sections from the FFPE intestinal tissues and stained with Luna, HE and TB (pH <2). And as described by Vogel et al. 2018, we also fail to differentially stain the mast cells in the small intestinal tissue using Luna and HE staining protocols. But since mMCP-4 is restricted to expression in CTMCs we therefore focused on the characterization of the submucosal mast cells. In TB stained (pH <2) mature adult intestinal tissues from non-infected mice and day 8 infected mice the counting of CTMCs in the intestinal lamina propria close to the smooth muscle layer are shown in new figure 2. This shows an increased level of submucosal mast cells in the intestinal epithelium during Giardia infections. This data and comments are now found in the paper.

Round 2

Reviewer 2 Report

The authors unfortunately did not experimentally address significant concerns raised in the previous review.

1-For example, this reviewer does not interpret the observation that female mice significantly loose weight up until day 4, unlike male congeners; he merely states the fact based on results shown in Figure 1 and reaches to the authors to provide an explanation. This is an intriguing observation and no interpretation is provided by the authors on that result.

2-Beyond monitoring IL-6 and providing convincing evidence that repression of mMCP-4 modulates the protein levels of that particular cytokine in Giardia instetinalis (GI) Infection in the mouse model, this reviewer suggests that the authors can not state in their discussion that :'... the mast cell  specific chymase mMCP-4 may play a role both in early and late host defence-mechanisms...' .  This reviewer would limit the interpretation to the impact of mMCP-4 in IL-6-dependent-host defence mechanisms.

3- Prior art indeed shows that mucosal mouse mast cells do not produce mMCP-4. The peritoneal lavage experiments requested by this reviewer aimed to answer three questions: i) would changes in peritoneal mouse mast cell counts in mMCP-4 KO mice play a role in the extent of GI infection  ii) are connective tissue mast cells, the main cellular source of mMCP-4 in the mouse GI infection model and iii) is there a cross talk between submucosal and connective tissue mast cells in GI infection. 

This third concern is reinforced by the fact that the authors did not attempt to monitor chymase (or mMCP-4 specific) activity in tissues extracted from naive or GI infected WT mice.
